

# Can katabatic winds directly force retreat of Greenland outlet glaciers? Hypothesis test on Helheim Glacier in Sermilik Fjord.

Iain Wheel[1, 2], Poul Christoffersen[1], Sebastian H. Mernild[3, 4, 5, 6]

[1]Scott Polar Research Institute, University of Cambridge, Cambridge, UK
[2]Department of Geography and Sustainable Development, University of St Andrews, St Andrews, UK
[3]Nansen Environmental and Remote Sensing Center, Bergen, Norway
[4]Geophysical Institute, University of Bergen, Norway
[5]Faculty of Engineering and Science, Western Norway University of Applied Sciences, Sogndal, Norway
[6]Antarctic and Sub-Antarctic Program, University de Magallanes, Punta Arenas, Chile

*Correspondence to*: Iain Wheel (iw43@st-andrews.ac.uk)

**Abstract.** Katabatic winds drive sea ice export from glaciated fjords across Greenland and other high latitude environments, but few studies have investigated the extent to which they also drive inflow of warm water and whether they have a direct impact on glaciers stability. Using ERA5 reanalysis data, verified by two local weather stations, we create a timeseries of

katabatic winds across Sermilik Fjord in southeast Greenland. Using this along with hydrographic data, from 2009-2013, positioned across the fjord, we analyse changes in fjord circulation during individual katabatic flows. Changes in melange presence are analysed too, via the use of MODIS and Landsat-7 satellite imagery. We show that warm water influxes are associated with katabatic winds, and that the potential submarine melt rates vary up to four-fold, dependant on katabatic wind strength. Rapid retreat of Helheim Glacier occurred during strong downslope wind events which removed the ice melange,

and so the well documented retreat of Helheim between 2001-2005 is predicted to be in part because of strong katabatic winds. Removal of the ice-melange led to a series of calving events, driven by a lack of buttressing and weakness propagation up the glacier causing a retreat of up to 1.5km. In contrast to previous research in which katabatic winds were seen as having an indirect influence on glaciers, we report direct forcing on Helheim Glacier through episodes of retreat occurring in response to inflow of warm water masses and removal of proglacial ice melange after downslope wind events.

**1 Introduction:**

The net mass loss from the Greenland Ice Sheet has increased from 51±17 Gt/yr between 1980-1990 to 286±20 Gt/yr between 2010-2018 (Mouginot et al., 2019) On average, approximately 40% of this loss stems from ice discharged by calving on tidewater glaciers which interact with the ocean as well as the atmosphere, and respond sensitively to hydrographic changes (Holland et al., 2008; Cowton et al., 2018). The remaining 60% is tied to surface melting and runoff, which is routed to the

base of glaciers and discharged subglacially into fjords where glaciers terminate in the sea. While some glaciers in Greenland terminate on land,  tidewater glaciers drain 88% of the ice sheet (Rignot and Mouginot, 2012). For glaciers in the marine setting, the two components of mass loss are linked because discharge of cold and fresh meltwater into warm and saline fjord waters generate convective plumes which melt ice and undercut the terminus (Rignot et al., 2010, 2015; Slater et al., 2017; Schild et al., 2018), and thereby increase the rate of calving (Benn et al., 2007, 2017; O'Leary and Christoffersen, 2013).


Circulation in fjords control the access of oceanic heat to most glaciers in Greenland, along with corresponding export of fresh glacial meltwater to the shelf (Inall et al., 2014; Jackson and Straneo, 2016; Straneo et al., 2016). As such, changes in fjord circulation can have wide ranging impacts, both to the Greenland Ice Sheet and global circulation (Bamber et al., 2012; Straneo et al., 2016). However, research into fjord circulation has mainly focused on shelf forced inflows driven by barrier winds

(Harden et al., 2014; Jackson et al., 2014; Straneo and Cenedese, 2015; Fraser and Inall, 2018), which are recurring high



velocity air flows down the East Greenland Coast due to the Greenland Ice Sheet acting as a barrier against the large-scale synoptic air flow (Harden et al., 2011; Moore et al., 2015). While the associated coastal downwelling and wind regime initially cause colder hydrographic conditions inside fjords, cessation of the barrier wind leads to a subsequent inflow of deep and warm Atlantic water in what is often known as the intermediary circulation inside fjords (Straneo et al., 2010; Jackson et al.,

2014). Coastal winds also control the flow of Atlantic water across the continental shelf, from the Irminger Sea towards the fjords (Christoffersen et al., 2011). However, there has so far been little consideration for other potential sources of meteorological variability such as katabatic winds (Christoffersen et al., 2012; Spall et al., 2017), which occur when dense air formation from radiational cooling over the Greenland Ice Sheet accelerates down the steep slopes that define its topography (Heinemann, 1999; Klein et al., 2001; Bromwich et al., 2002).


Katabatic winds are Downslope Wind Events (DWEs) in which dense air mass descends, in our case from the high-altitude ice sheet interior at 3,000 m above sea level. In Greenland, these DWEs can reach a magnitude equivalent to a hurricane on the Saffir-Simpson damage-potential scale, reaching wind speeds up to 90m/s (Born and Böcher, 2001; Mernild et al., 2008). Due to their cold air mass nature, they can reduce *in situ* temperatures to -20°C (Born and Böcher, 2001). This type of air flow

is very common in East Greenland, especially across Sermilik and Kangerdlugssuaq fjords where the coastal topographic gradients are extremely pronounced (Klein et al., 2001; Bromwich et al., 2002; Oltmanns et al., 2014). Although katabatic winds are approximately half as frequent as barrier winds (Jackson et al., 2014; Oltmanns et al., 2014), they are typically more intense (Spall et al., 2017), with local wind speeds capable of causing serious damage to the surrounding infrastructure (Born and Böcher, 2001; Mernild et al., 2008).


Katabatic winds have been suggested to influence fjord circulation (Sutherland et al., 2014) by outflow in the upper layer of cold and fresh fjord water (Moffat, 2014; Spall et al., 2017) and driving a return inflow of deeper warmer waters from the shelf into the fjord. This direct inflow differs from the intermediary circulation by barrier winds, which, in contrast, initially drives cold surface water into the fjords when wind speeds increase (Jackson et al., 2014; Sutherland et al., 2014). While

research has shown that intermediary circulation also includes a compensatory inflow of deeper warm waters when the coastal barrier winds drop, the inflow of oceanic heat from katabatic winds is more direct. Katabatic winds can, in addition, form polynyas, which influence glacier dynamics by removal of the ice melange, which provides a buttressing force on the calving front when icebergs and bergy bits are trapped in sea ice (Christoffersen et al., 2012).

In order to improve our understanding of the role of katabatic winds within the marine-terminating glacier system, we characterise DWEs and the effect they have on hydrography in Sermilik Fjord where Helheim Glacier is located. With a well-documented history of strong katabatic winds (Oltmanns et al., 2014, 2015), a relative abundance of hydrographic data from Sermilik Fjord (Harden et al., 2014; Jackson et al., 2014; Straneo et al., 2016), and a detailed record of acceleration and retreat of Helheim Glacier (Nick et al., 2009; Enderlin et al., 2014; Mouginot et al., 2019), this location is especially well-suited for

our investigation. We present a reliable historical catalogue of katabatic winds, drawing from both meteorological and reanalysis data, and compare these to changes in fjord temperature and current velocities while also analysing changes in the proglacial ice melange, calving front positions and sea-ice concentrations.

## 2 Data and Methods

### 2.1 Meteorology and climatology

Two weather stations were used to provide meteorological data for Sermilik Fjord (Fig. 1). The first was the Danish Meteorological Institute station in Tasiilaq which provided data spanning from 1958-2018 (Cappelen, 2019). The second was





the University of Copenhagen's research station on the banks of Sermilik Fjord (Mernild et al., 2008), which provided a timeseries from June 1998 until August 2014. We also use the European Centre for Medium-Range Weather Forecasts' ERA5 reanalysis product, which provides hourly timeseries data from 1979-2018 (Olauson, 2018).


Katabatic DWEs were defined as down-fjord winds sustained above a specified wind speed threshold for at least three hours, which is similar to previous work (Oltmanns et al., 2014). If the wind speed dropped to less than half of the specified threshold in events containing two peaks of katabatic strength, they were counted as two separate unimodal events. If not, then the two peaks were classified as a single bimodal event. A wind direction interval was also set to separate the katabatic winds from barrier winds which also give rise to high wind speeds. As it is known that the estimated magnitude of downslope winds can vary between the three datasets (Oltmanns et al., 2014) the parameters were set slightly differently for each one, which is justified by the lower wind speeds seen at the coast compared to the fjord and the underestimation of gravity flows in climate reanalysis products relative to weather station records (Oltmanns et al., 2014, 2015). As such, DWEs captured by the Fjord Station will show a higher wind speed compared to its representation in ERA5 data. For the Fjord Station (FS), the wind speed threshold was set at 15m/s with no direction parameter, since no non-katabatic wind produces a wind speed above the threshold at this location (Oltmanns et al., 2014). The parameters set for the Danish Meteorological Institute (DMI) station, were a wind speed of 11m/s and a direction between 280-360°. For the ERA5 dataset, the wind speed threshold was 10m/s and the wind direction window was 250-005°. The lower threshold applied for ERA5 data results in katabatic DWE classifications gave a similar frequency of those identified in the DMI station record, which is located within the ERA5 grid cell. The ERA5 dataset was used for most of the analysis in this study as, in contrast to the two weather stations, it provided consistent hourly wind speeds with no periods of missing data. Its predecessor, ERA-Interim, is known to have successfully predicted katabatic wind events (Oltmanns et al., 2014) and accurately represented barrier winds along the Southeast Coast of Greenland (Harden et al., 2011). ERA5 corresponded well with the two weather station datasets, especially between 2009-2014 (Fig. 2), making it a reliable source to show katabatic wind events (Oltmanns et al., 2014). The ERA5 pixel covering the township of Tasiilaq and the DMI station (Fig. 1) corresponded best with the weather stations, so its values were used to represent wind speeds over Sermilik Fjord and shelf. Using the discussed parameters, a catalogue of DWEs was created, using ERA5 reanalysis, for Sermilik Fjord from 1979-2018. The creation of this catalogue allowed the filtering of hydrographic, sea-ice and satellite imaging for the periods associated with katabatic winds. Regression analysis was used to compare ERA5 katabatic wind speeds above 10m/s against DMI station data wind speeds for the same times as the two datasets cover the same geographic area.

**2.2 Hydrographic data**

Moored buoy data was obtained for across Sermilik Fjord from the National Oceanographic Database Center (NODC). Data for the following seasons was obtained: 2010-2011 (Straneo, 2015), 2011-2012 (Straneo et al., 2015a) and 2012-2013 (Straneo et al., 2015b). The site numbers were given to each buoy based on the year it was placed and its buoy number within the accession. The location of each site used in this study has been mapped onto Sermilik Fjord (Fig. 1).

115

Temperature profiles for each site was linearly interpolated in 10 m bins to give a better resolution of the water column, similar to the steps taken by Jackson *et al.* (2014). The profiles from the upward facing ADCPs were backwardly interpolated from the bottom of the profile up, since recordings from shallower depths were often missing because of icebergs (Jackson et al., 2014). Temperature and current velocity timeseries were created for three days either side of each DWE occurring during the buoy's lifespan, based off the ERA5 DWE catalogue. The most relevant and informative profiles have been presented in this study (Results).



Potential Submarine Melt Rates (SMRs) were calculated for each site with a temperature profile to provide a quantification of the water column heat content. Again, the ERA5 catalogue of DWEs was used to provide a time reference frame and SMRs were calculated across the duration of each DWE between 2009-2013. SMRs were derived from the quadratic method used in Jackson *et al.* (2014). Here, SMR is defined as:

$$SMR = \frac{(T-T_f)^2 - (\overline{T-T_f})^2}{(\overline{T-T_f})^2},$$ (1)

where $T$ is the average water column temperature and $T_f$ is the *in situ* freezing temperature. Calculation of $T_f$ was performed using a modified Newton-Raphson iteration based on temperature, pressure and salinity (McDougall and Wotherspoon, 2014). The saturation rate was assumed to be one. Further to this Percentage Deviation of SMR (PDS) was defined:

$$PDS = \frac{SMR - SMR_{sr}}{SMR_{sr}} \; x \; 100 \; ,$$ (2)

where $SMR$ is submarine melt rate and $SMR_{sr}$ is submarine melt rate prior to DWE. As the extent of the temperature profiles varied at every site, $T$ was not always calculated at the same depth. Instead a 50 m section as close to 250 m as possible was used as $T$. To see the variation between DWEs, PDS for the particular event was normalised against the mean PDS over all DWEs at each site so to remove the variability from location within the fjord:

$$PDS \; change = \frac{PDS_K}{\overline{PDS}},$$ (3)

where $PDS_K$ is percentage deviation in SMR during a particular DWE and $\overline{PDS}$ is the mean percentage deviation during DWEs at the same site.

**2.3 Satellite imagery**

Satellite imagery for each DWE across the MODIS aqua timeseries, 2002-2018, based on the ERA5 DWE catalogue, was analysed. MODIS Terra and Aqua true color, thermal, Terra 367 and Aqua 721 were used, but only Aqua 721 is presented in this study as it provided the best degree of distinction between the sea-ice and the snow-covered land. MODIS satellite imagery from Aqua 721 was used to observe Sermilik Fjord and Helheim Glacier providing daily coverage, allowing observation of developing changes in proglacial conditions before and after the relatively short lived DWEs. To supplement the MODIS imagery, Landsat-7 was used to provide a better spatial resolution of the glacier terminus but its poorer temporal resolution meant images could only be analysed every seven days.

**3 Results**

**3.1 Katabatic winds**

Analysing the full ERA5 dataset from 1979, DWEs in Tasiilaq ranged in duration from 3hrs, the duration minimum cut off, to 53hrs and had a mean length of 15hrs of which on average 14hrs were above the 10m/s threshold. The maximum wind speed 24m/s was considerably lower than the Fjord Station and DMI datasets, which had maxima of 36m/s and 54m/s respectably. As shown by the methods of filtering the data for katabatic wind events, the fjord station showed the consistently higher wind speeds followed by the DMI station than ERA5 data.

Over the course of the ERA5 timeseries, 237 downslope wind events (DWEs) were identified resulting in an average of 6.1 events per year. In comparison, the DMI station record showed 416 DWEs at rate of 7.3 per year. The fjord station showed a





similar occurrence to ERA5 dataset of 6.6 DWEs occurring on a yearly basis, giving a total count of 119 events. The datasets compare well in identifying similar numbers of katabatic winds per year (Fig. 2) once the thresholds had been applied. Since 2001, the correspondence was extremely pronounced (Fig. 2). There is no clear pattern in the timeseries of either an increase

or a decrease, although there is a large interannual variability (Fig. 2). DWEs mainly occur in non-summer months with the highest monthly means between October and March (Fig. 2). Again, there is good agreement between the datasets although summer event numbers are noticeably higher at the Fjord Station. February numbers were considerably lower in the ERA5 data (Fig. 2). The regression analysis showed a significant relationship between ERA5 katabatic wind speeds and the DMI wind speeds at the same time (p<0.01). At the two main ERA5 thresholds used in this study, 10 and 20m/s, the equivalent

DMI wind speed was 16.8m/s and 23.6m/s respectively.

**3.2 Fjord circulation and temperature changes**

Circulation within the fjord was dominated by periodic up- and down-fjord currents which transformed into shifting east and west coastal currents across the shelf. Within the fjord itself this association was more clearly defined with up-fjord currents below 250m and an opposite flow above (Fig. 3). Flow speeds of water peaked in the mid-fjord at 0.8m/s, roughly halving

towards the mouth. The water flow speeds on the shelf rarely exceeded 0.2m/s. Speeds in the upper-fjord were generally between 0.25m/s in either direction. A full set of current and temperature profiles across the fjord and shelf between 2009-2013 can be seen in Wheel (2019).

At the mid-fjord prior to the DWE, the upper layer of water, above 250m, flowed down-fjord at around 0.4m/s. This down-

fjord flow intensified during the wind event reaching up to 0.8m/s. The flow rate seems to increase almost immediately with little noticeable lag once the wind speeds increase (Fig. 3). Below 250m, in the second water layer, there was consistently up-fjord flow throughout the DWE, again increased current speeds coincide with wind stress intensification. The up-fjord flow was present prior to the first event in 2012, and this flow continued during the katabatic wind with some apparent strengthening in velocity. When the up-fjord current was not previously present, a small, short-lived pulse of up-fjord flow, of around 0.3m/s,

was seen for several hours during the peak of the katabatic wind. Bottom layer, 525-550m, currents remained at a much lower magnitude with no clear alteration during or after a DWE (Fig. 3).

Within 12hrs of the decline of the wind forcing, the upper waters alternate to an opposite flow that is up-fjord (Fig.3). The strength and duration of this inflow were roughly equal in magnitude to the previous down-fjord flow it replaced. For the

weaker of the two events in 2012, the change in flow was relatively gradual and seems to be the upward movement of the deep-water current (Fig. 3). The change in current direction was much more rapid for the stronger event of 2012, with only a couple of hours separating a strong, 0.8m/s, up-fjord flow with a similar down-fjord velocity. The origin of the up-fjord flow in the top layer seems less clear too.

The temperature profile of the fjord changed in relation to the katabatic wind, depending on the fjord depth and position of the buoy within the fjord. Generally, the thermocline became shallower during and after a DWE although this was not always the case (Fig. 4). The response period of the water column also varied with its location and depth. Temperature profiles were closely aligned to currents assuming up-fjord currents were made up of warm water masses and colder temperatures occurred in down-fjord currents (Figs. 3 and 4). It should also be noted that salinity changed with temperature in accordance to current

knowledge of fjord properties (Jackson and Straneo, 2016). Clear increases in temperature corresponding to katabatic winds were seen in the mid- and upper-fjord (Figs. 4 and 5). Most notable, and in closest proximity to the glacier front, temperature increases of up to 0.75°C were recorded in the surface waters of the upper-fjord with temperature increasing almost immediately in response to down-fjord wind forcing (Fig. 5). For all the events analysed a sharp temperature jump occurred,





usually an increase of ~0.5°C, as the DWE became established. The peak was maintained for 4-12hrs before a sharp decline

in temperature occurred, even if down-fjord wind speed was maintained. Subsequence peaks of a similar magnitude were common, especially if the down-fjord wind speed remained high (Fig. 5).

Potential submarine melt rates, showing water column heat content, varied greatly across the fjord and with each DWE (Fig. 6). Comparing the relative strength of the events in 2010 against the change in PDS, increased heat content within the fjord is

associated with increased wind strength (Fig. 6). Events 1, 2 and 5 which have the lowest wind speed, coincide with a lower than average change in potential submarine melt rates resulting from the DWE. Furthermore, during events 1 and 2 all the sites display water temperature 50% lower than the average increase, whilst event 4, at the other end of the spectrum with the highest wind speeds, resulted in an almost 100% increase at all the sites (Fig. 6). This represents an up to four-fold increase in potential submarine melt rates across the fjord depending on the strength of the katabatic wind event. The maximum wind speed of the

DWEs shows a better association with the change in PDS than the duration of the DWEs (Fig. 6).

### 3.3 Ice melange removal

Most DWE events removed sea ice from the fjord and shelf along with partially breaking-up the ice-melange present in front of the glacier terminus. For DWEs in which wind speeds did not exceed 20 m/s, only small fractions of the melange in front of Helheim Glacier broke up. However, over the subsequent few days after each event, we found sections of ice and icebergs

held within it were released down the fjord. Stronger events in which wind speed exceeds 20m/s did more damage, breaking-up the majority of the ice melange leaving just small sections in front of the terminus (Fig. 7). In the days following a strong DWE the glacier terminus is exposed to areas of open water or thin sea ice (Fig. 7). The results of a series of large calving events can be seen in the week after the strong event in March 2011 (Fig. 7) with large tabular full-thickness icebergs observed in front of the terminus (Fig. 8). Initially the calving occurred on the south side of the terminus where it was exposed to open

water, but subsequent calving took place on the northern side. Along with calving, as shown by the newly created icebergs, the terminus underwent noticeable retreat of up to 1.5km in the nine days after the katabatic wind event (Figs. 7 and 8).

A similar pattern was seen in 2005 and 2013 in the weeks after strong events following the break-up of the ice-melange. Unfortunately, these are the only other strong events within the MODIS data series due to clouds frequently obscuring the

glacier front in the imagery. Sea-ice and melange through all events observed showed strong rotational influences with sea-ice or melange removal occurring to the right of the wind direction (Fig. 7). We also found strong katabatic wind events to primarily cause retreat on the southern part of the calving front.

### 4 Discussion

#### 4.1 Katabatic winds

DWEs have been shown to be recurring events across Sermilik Fjord, with an annual occurrence of around seven, that show no discernible long term change over the last 60 years (Fig. 2; Oltmanns et al., 2014). The strength of each DWE event also varied independently over the duration of the timeseries (Fig. 2). However, the prevalence of DWEs was closely linked to seasonal changes, with the highest occurrence in winter when radiational cooling is highest (Fig. 2). Logically, it could be assumed that the decline in mass of the Greenland Ice Sheet would have reduced katabatic winds through a reduction in

radiational cooling, but this is not the case. Instead DWEs and strong DWEs do not show a long-term trend (Fig. 2), possibly because the steep topography of the region means little cooling is required in katabatic wind formation.
None of the wind events in our data were close to the potential maximum wind speed of 90m/s often referenced in the literature for extreme katabatic events (Born and Böcher, 2001; Oltmanns et al., 2014). We attribute this to the role of topography in





driving the maximum wind speed locally. More importantly, the maximum downslope wind speed observed in the ERA5 data,
24m/s, was well below the Fjord Station maximum of 36m/s - suggesting that ERA5 does not resolve the full strength of
katabatic winds as observed. A similar issue was discovered in ERA-Interim, which failed to resolve mountain wave breaking,
so it too underestimated downslope air flow speed, suggesting a comparable problem to that noted in ERA5 (Oltmanns et al.,
2014, 2015). It is possible that insufficient resolution of topography and the models grid cell size explain these shortcomings
(Oltmanns et al., 2015). Importantly, though, this study has shown that ERA5 underestimates downslope flow by 18±6.9% for
an ERA5 speed of 20m/s, for the katabatic wind events identified in this study, allowing the correction of this shortcoming in
future work. Despite these limitations, we found an overall good correspondence between katabatic events in ERA5 and
weather station data, which shows that the former can be used to identify katabatic winds in future work, e.g. on spatial scales
much larger than those offered by local weather station records (Fig. 2). This result suggests further analysis refining the
relationship between katabatic wind speed and ocean surface stress would be beneficial given the potential usage of ERA5 to
model ocean heat loss in the Irminger Sea (Josey et al., 2019).

**4.2 Fjord circulation and temperature changes**

Deep influxes of warm Atlantic origin water are seen throughout Sermilik Fjord as a result of katabatic winds, and this inflow
usually persists throughout the DWE (Fig. 3). Current profiles in the mid-fjord indicate a pycnocline between the two currents
of opposite directions at around 250m prior to and during DWEs (Fig. 3), agreeing well with previous studies in Sermilik
(Straneo et al., 2012; Jackson et al., 2014). During a DWEs currents above 250m flow down-fjord, while those below flow up-
fjord before wind cessation causes a directional change to both currents. Specifically, currents above 250m switch to up-fjord
flow and those below switch to down-fjord flow (Fig. 3).

During DWEs, warm bottom water inflow is balanced by equivalent outflow of cold glacially modified water near the surface.
This circulation regime differs from the intermediary fjord circulation driven by barrier winds because the inflow of warm
Atlantic water takes place during the event itself and not afterwards. Furthermore, a sudden alteration in the direction of the
current occurs immediately after the Katabatic wind stress is reduced. Hence circulatory changes within the fjord during and
after katabatic DWEs seem to be almost immediate (Fig. 3). This trend is also shown in the temperature profiles within the
fjord and on the shelf, and the short response time to the wind forcing seems to be roughly uniform no matter what the location
(Figs. 4 and 5). These circulatory changes are similar to the final stage of intermediary circulation (Straneo et al., 2010; Jackson
et al., 2014; Sutherland et al., 2014; Spall et al., 2017), but without the major time delay following the wind forcing (Harden
et al., 2014). The immediate increase in temperatures (Figs. 4 and 5) and warm inflow throughout the fjord means katabatic
wind driven circulation must be highly efficient at transferring heat up the fjord.

Another difference between katabatic wind driven circulation relative to intermediary circulation concerns the increases in
temperature of surface waters near to the front of the ice melange (Fig. 5). Intermediary circulation has been shown to greatly
increase the heat content within the fjord, but it remains unclear how effective such circulation is at heating the upper fjord,
close to the glacier terminus (Jackson et al., 2014). Although modelling studies in Kangerdlugssuaq have shown intermediary
circulation to be highly efficient (Fraser and Inall, 2018), it would not be expected to heat surface waters but rather to increase
water column heat content through shoaling of the thermocline following the relaxation of the wind set up in the days following
(Straneo and Cenedese, 2015). Temperature increases at the top of the fjord can be explained by deep water upwelling as
surface waters are exported down the fjord from the katabatic wind stress (Fig. 5). Since this process is occurring in the upper-
fjord, it can be assumed a similar process is occurring at the glacier terminus itself potentially causing other melting processes
such as free convection (Cowton et al., 2016; Schild et al., 2018). If this mechanism occurs, the glacier terminus could be
exposed to warm waters at all depths. This could lead to rapidly increasing melting when ambient fjord water is entrained in





convective plumes driven by cold and fresh subglacial discharge (Jenkins, 2011). As such, the relative heat transport into the fjord from the shelf is less significant as our results show that katabatic winds can drive warm water from within the fjord itself to the glacier front (Fig. 5).

Although the short-lived temperature increases we observe may mean that katabatic wind-driven shelf-fjord exchanges are less significant compared to those driven by the barrier winds (Jackson et al., 2014; Spall et al., 2017), the latter occur on the shelf and near fjord mouths, which are often far from the termini of glaciers (Straneo and Cenedese, 2015). In contrast, katabatic winds often reach their maximum where the ice sheet terminates. Hence, even if barrier winds have a high annual frequency and longer duration (Oltmanns et al., 2014; Spall et al., 2017), we hypothesise that submarine melting caused by

katabatic winds could be equivalent given that maximum wind stress often occurs deep inside fjords near the termini of tidewater glaciers. Furthermore, since katabatic driven melt at the terminus is caused by upwelling of fjord waters, barrier wind driven renewal events could aid this process by increasing bottom water temperature. Our results indicate that future modelling studies should consider katabatic wind driven changes and that shelf-fjord heat exchange is not equivalent to heat changes at the glacier.


The calculated potential SMRs clearly show heat content within the whole fjord increases with increased katabatic wind strength, once seasonality and position within the fjord are accounted for (Fig. 6). Changes in potential submarine melt rate changed by at least four-fold across all fjord sites between strong and weak DWEs (Fig. 6). Maximum katabatic wind speed is a much better predictor of increased water column heat content or SMR than the duration of the DWE, suggesting a dominant

effect of wind speed over duration in how wind stress influences hydrographic fjord properties. Currently, model based studies of katabatic wind influences do not account for the large variations in wind speed (Spall et al., 2017; Spall and Pedlosky, 2018). This may be a limitation given that wind stress is a quadratic function of wind speed (Large and Pond, 1981). Any variation unaccounted for in wind speed may be magnified in SMR or water column heat content estimations as an increase in wind speed of 10m/s, <80%, led to a four-fold, 400%, increase in melt rates (Fig. 6). The influence of katabatic winds on

glaciers could therefore be significantly greater than previous studies have suggested.

### 4.3 Ice melange removal and calving

Above, we discussed how submarine melting caused by inflow of deep Atlantic water in response to katabatic DWEs should lead to undercutting of glaciers. This process should, in theory, result in faster calving and potentially significant rates of terminus retreat (O'Leary and Christoffersen, 2013). The glacier terminus may in addition be weakened further by removal of

the buttressing force acting against the terminus when icebergs and bergy bits form a rigid melange in front of glaciers in winter. Previous studies have showed that significant retreat of Kangerdlussuaq Glacier farther north in East Greenland coincided with a wind-driven removal of rigid proglacial ice melange (Christoffersen et al., 2012), and this link is supported by numerical models showing a sensitive response of calving glaciers to changes in the buttressing force (Todd and Christoffersen, 2014; Todd et al., 2018, 2019). Hence, we analysed how the proglacial melange in front of Helheim Glacier

was affected by DWEs.

Our study shows that a typical DWE event created open water within the upper-fjord, which eroded sea-ice near the terminus in the days following. Simultaneous temperature spikes are short lived, often around 8hrs, so although there is some evidence of melting occurring by the rapid reduction in temperature and steadily maintained salinity (Fig. 5), melting cannot be

responsible for the subsequent sea ice break up. Instead katabatic wind stress is likely to have caused fracturing of the sea-ice and aided by rising air temperature following the DWE facilitates the melting of the broken sea ice. While the observed DWEs did not always create open water in front of Helheim Glacier, we found those in which wind speeds were higher than 20 m/s





to consistently do so, and with open water replacing rigid melange we observed increased calving and frontal retreats in the week following these events (Figs. 7 and 8). Melange and sea-ice removal occurs during the DWE or within a few days after as fjord currents remove the fractured segments (Fig. 7). The open water observed in the southern side of the fjord in the proglacial fjord is telling, as sea-ice removal was found to a show clear rotational influence (Fig. 7). Down-fjord wind stress driven Ekman transport must lead to wind set up along the western side of the fjord, which in turn drives a strong down-fjord current, causing sea-ice removal to predominate here (Fig. 7; Oltmanns et al., 2014). We hypothesise a method by which sea-ice is fractured by wind stress, then removed by down-fjord currents resulting in the across fjord variability in sea-ice removal (Fig. 7). If wind driven fracture did not occur, across fjord variability would be far less pronounced. This again would suggest open water formation at the top of the fjord is driven by wind fracture aided by wind driven circulation changes, rather than melting or circulation changes as the primary driver.

Given that warmer winters are causing lower sea ice concentrations due to climate change (Christoffersen et al., 2012), there is a realistic possibility that DWEs will export sea ice more easily compared to present conditions, leading to more open water near glacier termini – even if the frequency of DWEs remains unchanged. Interestingly, we found the melange is not necessarily removed during the DWE but instead disintegrated over the following days, suggesting again a wind stress induced weakening and fracture-driven break-up. Similarly, to the reasoning above, short lived temperature changes are unlikely to cause such a pronounced break-up meaning a strong wind stress threshold, hypothesised to be 20m/s on ERA5, which translates into an infield measurement of 23.6±1.38m/s based off the regression analysis.

### 4.4 Terminus stability

Katabatic winds have been shown to correlate with the retreat of Kangerdlugssuaq Glacier between 2004-2005, along with air temperature anomalies (Christoffersen et al., 2012). Our results show strong katabatic winds can alter the terminus position of Helheim Glacier by up to 1.5km in little over a week (Fig. 8), through ice-melange removal and increased SMRs. Although there seems to be no long-term trend in katabatic winds, a number of strong katabatic winds occurred in 2004 and 2005 (Fig. 2). Hence it is feasible that strong DWEs triggered or sustained the rapid of Helheim Glacier in 2004-2005 (Howat et al., 2005; Luckman et al., 2006; Miles et al., 2016). To explore this possibility further, we provide an example of the influence of DWEs by analysing frontal positions of Helheim Glacier in a katabatic event in our DWE catalogue. Following the disintegration of the ice melange following the DWE on 14/03/2011, the terminus underwent significant retreat of up to 1.5km and calving, occurring 3-12 days after the initial DWE (Figs. 7 and 8). Apart from this presented example, the other two instances of strong DWEs visible within the MODIS timeseries both showed retreat. One underwent significant calving for around five days, whilst the other showed noticeable but less pronounced retreat and no calving of large tabular icebergs. The sustained calving of the glacier in the days after the event could mean a shift in the internal calving dynamics (Benn et al., 2007; Bassis and Jacobs, 2013) caused by katabatic winds. While this influence could either be indirectly through increased submarine melting or directly via the removal of the buttressing provided by the ice melange, the short-lived nature of the temperature increases (Fig. 5) and the absence of surface melt water as a driver of plume-induced undercutting in winter months with no surface ablation suggests that the latter is more likely.

Removal of the ice melange is known to provide important buttressing to the terminus but studies have mainly focused on spring and summer months (Amundson et al., 2010; Christoffersen et al., 2012)- although it has been speculated as an important factor in winter readvancement (Todd and Christoffersen, 2014; Todd et al., 2018). Modelling studies of Store Glacier in West Greenland suggest that rigid melange forming in winter months promote terminus advance by suppressing crevasse propagation and calving and that significant retreats occur early in the melt season when warmer air temperatures melt sea ice and allow the melange to break up (Todd et al., 2018, 2019), mirroring the theorised pattern at Helheim with the exception





that katabatic winds can also remove the melange temporarily in the middle of winter. It should also be noted that no noticeable retreat was observed in weaker events that failed to remove the ice melange, backing up the theory that melange buttressing may be the dominant control on calving dynamics (Reeh et al., 2001; Christoffersen et al., 2012; Todd and Christoffersen, 2014).

The sustained retreat and calving in the strong event in 14/03/2011 (Figs. 7 and 8) suggests a sensitive response in calving, which was originally suggested to be an amplification in the calving rate due to intensification of extensional stresses when the ice front is undercut by submarine melting (O'Leary and Christoffersen, 2013; Benn et al., 2017). The loss of a buttressing force from ice melange removal could result in a similar instability because the force is centred near the water line. The position of the buttressing force effectively reduces the torque and bending moment at the calving front where there is an imbalance of

hydrostatic forces (Reeh, 1968; Benn et al., 2007). As such, the rapid retreat of up to 1.5km could feasibly be a result of a torque and bending moment induced by removal of melange, with the result that calving is amplified.

## 5 Conclusions

We studied katabatic flows in Sermilik Fjord using observational station data and the ERA5 climate reanalysis product. On average we found katabatic DWEs to occur 6.6 times per annum with wind speeds of up to 36m/s in the Fjord Station data,

although lower (24m/s) in ERA5 reanalysis data. The annual occurrence was nevertheless very similar at 6.1 per annum and no long-term trend was observed. While the wind speeds of DWEs in ERA5 was 18% lower than observed at the DMI station, we found the reanalysis to broadly capture the correct occurrence of these high-intensity events with the application of suitable wind speed threshold. By examining hydrographic data, we found katabatic winds cause significant changes in the circulation across Sermilik Fjord, with warm water drawn into the fjord from the shelf (Figs. 3 and 4) but more significantly deep warm

water in the fjord upwells and increases temperatures in upper waters near the terminus of Helheim Glacier (Fig. 5). While the circulation of water in response to katabatic winds is less sustained compared to those reported previously from barrier winds (Straneo et al., 2010; Jackson et al., 2014; Cowton et al., 2016; Fraser and Inall, 2018), they are more direct in the sense that the along-fjord katabatic winds drive compensatory inflow of warm Atlantic water at depth when the wind stress is high. For barrier winds, this inflow is indirect as it only occurs once the winds die down (Straneo et al., 2010; Sutherland et al., 2014).

We also found katabatic winds to be driving a heat exchange into the fjord from the shelf (Figs. 3 and 4), although it is unclear whether this exchange is strong enough to drive transport heat from the fjord mouth to the top of the fjord in a single event. Ultimately, the fjord-shelf heat exchange at the fjord mouth is likely to be driven by the more sustained along-coast barrier winds (Straneo et al., 2010; Jackson et al., 2014; Sutherland et al., 2014; Fraser and Inall, 2018).

We found DWE strength to be directly related to potential submarine melt rates (Fig. 6), which means the inflow of warm water may be sufficiently high to undercut glaciers. In fact, it is feasible that strong katabatic winds in the early 2000s (Fig. 2) influenced the abrupt retreat of Helheim Glacier during this period of time (Howat et al., 2005). This sensitive response may, however, at least not solely depend on circulatory inflow of warm water because the inflow driven by DWEs and barrier winds tend to occur in winter when there is an absence of glacial runoff to drive high submarine melt rates by entrainment of fjord

waters into buoyant plumes (Jenkins, 2011).

Our study corroborates earlier work which found a potentially sensitive response of marine-terminating glaciers to removal of rigid proglacial ice melange by DWEs (Fig. 7 and 8; Christoffersen et al., 2012). This response may be especially important for fast-flowing glaciers such as Helheim, which bring ice to their terminus at a faster rate than the terminus can melt due to

contact with seawater (Todd et al., 2018, 2019; Cook et al., 2020). We found melange at Helheim to be removed when katabatic





wind speed in the fjord is greater than 20m/s (Fig. 7). As a result, this study shows how DWEs can remove the buttressing force from the ice melange during non-summer months when calving activity is typically suppressed (Amundson et al., 2010; Todd and Christoffersen, 2014; Todd et al., 2019). Katabatic winds could have therefore been a factor in the change in calving dynamics that led to a rapid retreat of Helheim Glacier (Fig. 8, Howat et al., 2005) as well as Kangerdlugssuaq Glacier farther

north (Christoffersen et al., 2012). This direct role of winds stands in contrast to previous studies, which found fjord circulation to be the dominant driver of tidewater glacier dynamics (Straneo et al., 2010; Inall et al., 2014; Jackson et al., 2014; Straneo and Cenedese, 2015).

While there is no trend in the number of DWEs per year in our 60 year record from Sermilik Fjord (Fig. 2; Oltmanns *et al.*

2014), the could still be long term trends in elsewhere in Greenland. The effect of katabatic winds on glaciers through removal of ice melange may, regardless, still become more pronounced given that winter air temperatures are rising faster than summer air temperatures and sea-ice concentrations are dropping across the Arctic as a whole. While we found only strong events with wind speed higher than 20 m/s to consistently break up the melange in front of Helheim Glacier, this apparent threshold in wind speed is likely to reduce over time in response to thinner sea ice cover and weaker melange forming in front of

Greenland's marine-terminating outlet glaciers. Hence, we hypothesise that the importance of katabatic winds will increase in the coming decades and century.

*Acknowledgements:* IW acknowledges support from the Scott Polar Research Institute through the Debenham Scholarship. PC acknowledges support from the Natural Environment Research Council (grant no. NE/K005871/1) and the RESPONDER grant awarded from European Research Council through the European Union's Horizon 2020 Research and Innovation

Programme (grant no. 683043).

*Data availability:* All the data used in this paper is freely available online. ERA5 reanalysis data is available at: https://cds.climate.copernicus.eu. The Danish Meteorological Institute Tasiilaq data is available at: https://www.dmi.dk/publikationer/. MODIS imagery is available at: https://www.ncdc.noaa.gov/gibbs/. Landsat 7 imagery is available at: https://earthexplorer.usgs.gov/. Hydrographic buoy data from Sermilik Fjord is available at:

https://www.nodc.noaa.gov/, accession numbers 0123282, 0126772 and 0127325.

*Author Contributions:* IW designed the study and conducted the analysis with guidance from PC. IW wrote the manuscript with significant input from PC. SM provided the Fjord Station data.

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





**Figure 1:** Location map showing Sermilik Fjord and Helheim Glacier in southeast Greenland. Coloured triangles and dots show locations where air and water temperatures were recorded by weather stations and buoys respectively. Square shows location of ADCP current velocity measurements. Site 11_5 has the exact same location as site 10_4. The ERA5 pixel used shown as a hatched box. Only locations of buoys used in this study are shown. Inset shows a regional map produced with Esri world imagery.




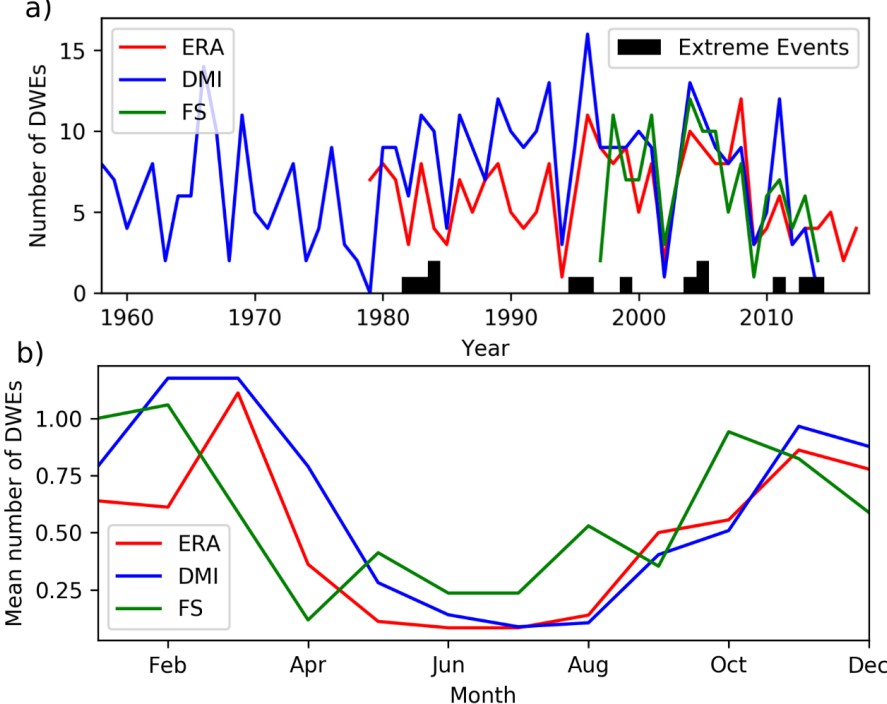

**Figure 2: a)** Katabatic DWE events in Semilik Fjord as identified in ERA5 reanalysis and two station records. Black bars represent DWEs over 20m/s based on ERA5 data only. **b)** comparison of mean number of DWEs per month throughout the year. The DMI station in Tasiilaq spans from 1956-2014. The Fjord Station record from Sermilik Fjord spans from Jun 1997 to Aug 2014. The ERA5 reanalysis data are for Tasiilaq and spans from 1979 to 2018.



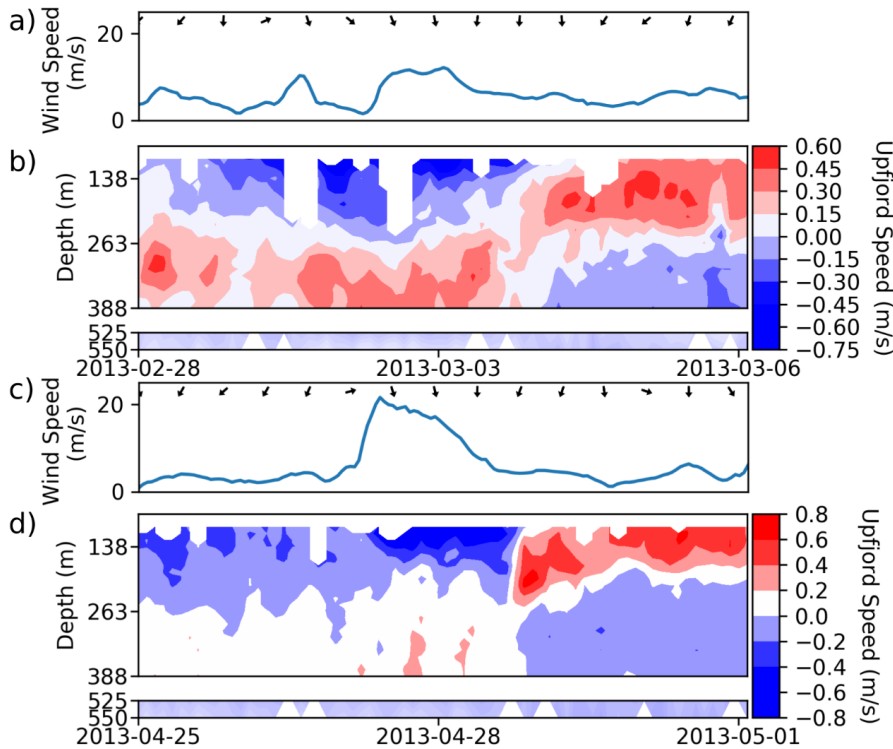

**Figure 3: Wind speed and direction for two DWEs in Sermilik Fjord in 2013 (a, c) and the corresponding current velocities observed in the plane of the fjord direction from buoys site 12_4 halfway up the fjord (b, d). Wind speed and wind direction are from the ERA5 data. See Fig. 1 for exact location of ERA5 pixel and buoys. Note the differing speed scales between the panels. The second DWE is also shown in Fig. 4 g-h.**








**Figure 4: Wind speed and direction for four DWEs in Sermilik Fjord (a, c, e, g) and the corresponding temperature profiles from buoys site 10_2 near the middle of the eastern side of the fjord (b, d, f) and buoy site 12_4 at the middle of the fjord (h). Upper fjord temperature and salinity changes during the three events in 2011 are shown in Fig. 5, whilst current velocities during the DWE in**





2013 is shown in Fig. 3. Wind speed and directions are from the ERA5 data. See Fig. 1 for exact location of ERA5 pixel and buoys. Note the differing temperature scales between the panels.

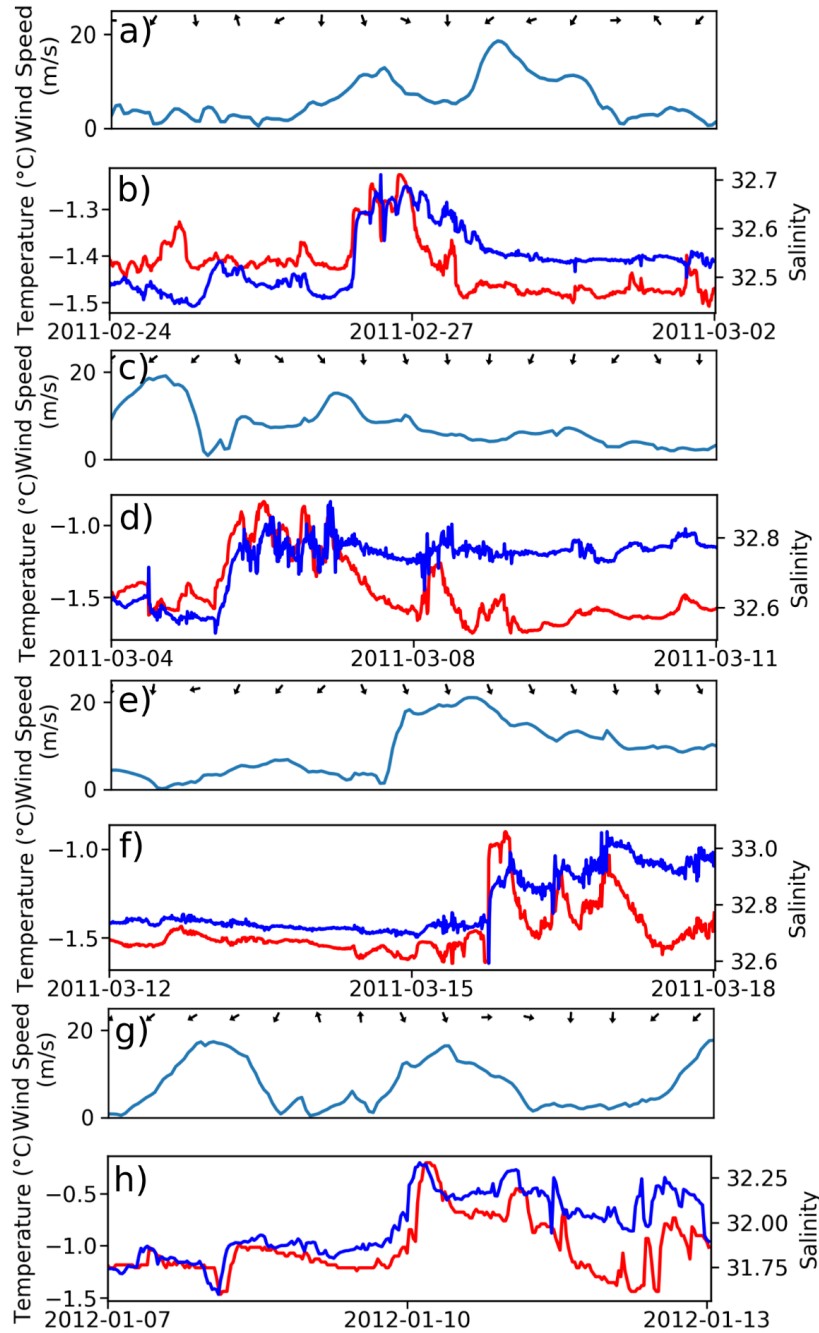

**Figure 5: Wind speed and direction of four DWEs observed in Sermilik Fjord from the ERA5 data between 2011 and 2012 and the corresponding observations of temperature (red) and salinity (blue) recorded by buoys 10_4 and 11_5 in the upper-fjord near Helheim Glacier at a depth of 13m. Wind speed and direction are from ERA5 data. Hydrographic changes during the events in winter of 2010 are shown in Fig. 4. See Fig. 1 for exact location of ERA5 pixel and buoys. Note the differing scale between the panels.**





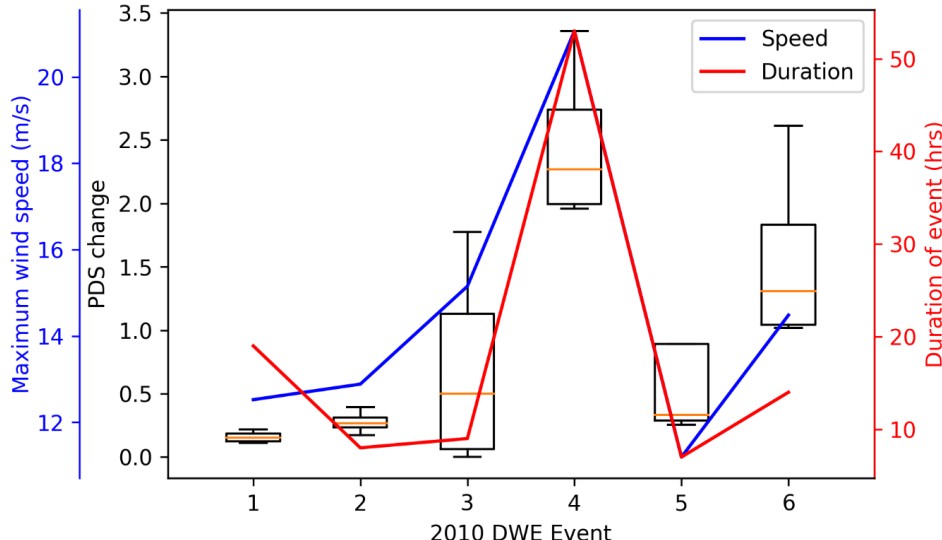


**Figure 6: Boxplots showing the change in Percentage Deviation of Submarine melt rates (PDS, left axis) across Sermilik Fjord. Each boxplot represents the change in PDS during the six DWEs that occurred in 2010 season, for all the 2010 sites shown in Fig. 1, which have been normalised against the mean change in PDS at each site across all six DWEs. The temperature profile changes in events 2, 3 and 4 are also presented in Fig. 4 and 5. The red line shows the duration of each DWE (right axis), whilst the blue (left axis)**

**shows the maximum wind speed.**



**Figure 7: a)** MODIS Aqua 721 image of Sermilik Fjord and showing the removal of sea ice. **b-g)** MODIS Aqua 721 images of Helheim Glacier and the top of Sermilik Fjord showing the retreat of the glacier terminus after a strong DWE that started on 14/03/2011 with maximum wind speed of 21.1 m/s and duration of 53 hrs in fjord station data. Solid red line marks the terminus position on 17/3/2011. **a)** shows open water in front of Helheim Glacier after the DWE while **b-g)** shows how the front of the glacier subsequently retreated due to loss of buttressing from melange and sea ice. Large tabular icebergs can be seen floating away from the terminus. Higher spatial resolution images of this event are shown in Fig. 8. The changes in fjord temperature associated with this event are presented in Figs. 4 and 5.



**Figure 8: Landsat 7 false colour images of Helheim Glacier and the top of Sermilik Fjord showing the retreat of the glacier after a strong DWE that started on 14/03/2011 which had a maximum speed of 21.1m/s and duration of 53hrs in fjord station data. a) Image taken on 16/03/2011 with solid red line showing the terminus position and dashed red line a large crevasse. b) image taken on 23/03/2011. Frontal changes in higher temporal resolution is shown in Fig. 7 and contemporaneous fjord temperatures can be seen in Figs. 4 and 5. The stripes of missing data is caused by a scan line error in the Landsat 7 satellite.**