# Peer review of "Can katabatic winds directly force retreat of Greenland outlet glaciers? Hypothesis test on Helheim Glacier in Sermilik Fjord."

_The Cryosphere, 2020_

## Referee Comment (RC1) · Anonymous Referee #1 · 19 Aug 2020

General comments:

Wheel et al. present their findings of how katabatic winds draining off the southeast Greenland ice sheet influence Sermilik fjord circulation and sea ice/ice melange break up in front of Helheim Glacier. A catalogue or climatology of katabatic winds, categorised into Downslope Wind Events (DWEs), is created using ERA5 data and compared to two weather stations. They use a combination of atmospheric and ocean observations with satellite products to investigate a number of events in more detail and provide convincing evidence that katabatic winds have a direct impact on the fjord circulation. Furthermore, they provide evidence for glacier retreat and calving directly

caused by the katabatic winds in a particularly strong event.

The combination of atmosphere and oceanographic observations over a number of years highlights the results in a comprehensive way. Some of their results are interesting and do add to the research and knowledge in this field, however some are not novel and look very similar to the findings of Oltmanns et al. 2014. The discussion is thorough and well balanced, with them highlighting the new research and the gaps that remain. The limitations of the study should be made clearer though, and the motivation of the study should be more obvious in the introduction.

My reservations lie in four main areas, two of which the authors could remedy without too much additional analysis. However, two may require more work. These are described more below. The research fits well in The Cryosphere journal and will be useful for the atmosphere and coastal oceanography community. The authors mention a number of areas that future research could go into, based on their results. If the major and minor comments are addressed, then it should be accepted for publication.

Firstly, the more major changes:

Some of your results, specifically the catalogue of DWEs is not novel, as it is almost identical to the Oltmanns et al 2014 study. In Oltmanns et al. 2014, they use the same two weather stations as you, use the same method but with slightly different thresholds, and present the findings using the same figures. However, you fail to mention that your study uses the Oltmanns method, with some changes. You also do not mention the Oltmanns study in the introduction- which you could use to your advantage, as there were gaps remaining in this research field, and your findings fill these gaps by building on the Oltmanns et al. 2014 foundation. There is also little comparison to Oltmanns et al. 2014 in your results or discussion, even though their results are relevant, as they use the same data as you, and also created a catalogue of katabatic winds. For instance, Figure 2 in your manuscript is the same as Figure 3 in Oltmanns et al. 2014, therefore this is not a new finding, but more of an extension of the Oltmanns study but

using ERA5 data. You need to make clear in the figure caption that it is an adaptation of the Oltmanns et al 2014 one. Similarly, Oltmanns et al. 2014 also looked at the impact of the March 2011 katabatic event on sea ice cover (Figure 10 in Oltmanns et al. 2014), which is similar to your finding in Figure 7. This means that two aspects of your results are not quite as novel as they could be (catalogue of DWEs and March 2011 sea ice analysis). The Oltmanns et al. 2014 study didn't include any hydrographic data or discussion on fjord circulation, so this is where your research comes in, but that isn't clear enough currently. You should strengthen your motivation for the study in the introduction. If you: 1) explain that you use the Oltmanns method but alter it for X,Y,Z reasons, 2) explain that you have used Oltmanns study as a basis but updated it for ERA5, 3) discuss the ERA-Interim/ERA5 differences in the discussion, 4) describe why you also look at the March 2011 episode (e.g improved resolution of satellite imagery provides more information about ice melange break up and movement of ice bergs), then I think this manuscript could be improved.

What is the evidence for sea ice break-up related to katabatic winds if satellite imagery is only available (due to cloud cover) for three events? As there are only three events, and the March 2011 one has previously been published, why didn't you present one or both of the other events? This would be more novel and provide stronger evidence for your conclusions that katabatic winds influence sea ice concentration. Currently, you make quite bold statements 'most DWE events removed sea ice from the fjord…' (line 212) but there isn't much evidence to back up this statement. You show evidence of this from March 2011, which has been shown previously, and say that you only have imagery from two other events. So how do you know that most events remove the sea ice?

Secondly, the two minor changes:

The black bars in figure 2 and results/discussion surrounding the extreme events are not included in the results. Similarly, the 20m/s threshold of 'extreme' events is not introduced in the methodology. Currently, the reader can gain very little about where

this 20m/s threshold came from, and what can be interpreted from this. Only further into the results, where the sea ice break-up is presented, is the 20m/s value introduced. This should come earlier in the methodology and in the first section of the results where you present the catalogue of DWEs.

Section 2.3 of the results needs restructuring to make it clearer where the authors are presenting the general characteristics of the DWEs, or where they are pointing directly to the case studies. This part of the manuscript needs improvement, as I have made many smaller/specific comments below also. The results are robust and interesting, but they are currently difficult to interpret, and it took me a number of reads through the manuscript to understand. The specific comments relate largely to confusion over dates in the text and figures (which differ) and highlighting specific periods in the figures for ease of understanding. For example, adding dashed lines to show when the DWE starts and ends. Citing specific panels of figures (e.g Fig 3c instead of Fig 3) might make this section easier to understand also. Please see the specific comments below for other suggestions.

Specific comments:

Intro, Ln 27-30: What is the citation for this 40% calving, 60% surface melt statistic?

Intro, Ln 54: reorganise sentence to make it clearer. For example: 'As the katabatic air mass is colder than ambient conditions, they can reduce in situ temperatures ...'. With the -20°C, do you mean that the airmass reduces temperatures by 20°C (i.e a change) or that absolute temperatures can reach -20°C? This isn't clear.

Data/Methods: In your methodology you do not say that the method you use is taken from Oltmanns et al. 2014. The citations of the study are not enough, as it is not made clear to the reader that this is not a new method, but slightly adapted values from a previous study. You need to make this clearer and give justification for using different values if a previous catalogue is available.

Ln 80-85: More information about the AWSs and ERA5 is needed. What is observed at these stations and what data do you use? E.g. Wind direction and wind speed only. At what interval are the observations made and what interval do you use? Introduce your abbreviations here too, so that it is consistent throughout the manuscript. What resolution is ERA5? Did you select one grid point for analysis or an area average? Cite Figure 1 so that readers can see the ERA5 location.

Ln 84: The Olauson 2018 citation is not necessary here, as they looked at ERA5 for wind power in different countries and is not related to this study. Use a citation provided by ECMWF or their doi for this.

Ln 104: The Oltmanns et al. 2014 citation isn't needed here, as that study using ERA Interim, not ERA5, so did not show its reliability.

Ln 125 and 132: What time period prior to the DWE do you consider for this calculation? Is it a certain number of hours/days/weeks? As you only mention the 'duration' of the DWE on ln 125.

Ln 140: Later in the manuscript, you say that due to cloudy conditions, you can only assess ice break up for specific events in 2005, 2011 and 2013. Please give an indication of this lack of data in this paragraph. As it reads currently, it sounds like you have a daily timeseries of such data.

Results, Ln 155-157: remind the reader of the time period of each data set here. E.g '. . . the DMI station record (1958-2018) showed. . .'.

Ln 164: This is the first time that 20m/s has been mentioned as a threshold, so it confused me that it was mentioned here. I see on Figure 2 that this corresponds to an extreme event, but this is not explained in the data/methods section, or elsewhere in the results. Please put the black bars from Figure 2 into context in the results section. It becomes more important later on with the discussion of sea ice breakup, but I only know that from reading the manuscript a few times.

line 176: flow rate increases with little lag, then Figure 3 is cited. Is this a general comment, or only specific to the events shown in Figure 3?

Figure 3/4/5 would also benefit from a dashed vertical line of some kind to highlight when the DWE starts/ends.

For fig 3a there are 2 peaks in WS- is this 1 DWE or 2? Some dates should be included in the results section to highlight which DWE you are talking about. E.g Line 185: 'the weaker of the two events'- which one does that relate to? Include a date and/or highlight the specific panel in Figure 3.

Ln 178: 'first event in 2012' but the dates in figure 3 are 2013. Same for line 185.

Ln 191: Does the figure 4 citation here relate to where the thermocline became shallower, or when it was 'not always the case'? Point to a specific panel if needed.

Ln 204: 'events in 2010': none of your figures are from 2010, so are you now looking at 2010, or should that be a different year?

Figure 4/5: There are some date issues on these figures. Figure 4c/d is 2011-03-05 whereas Figure 5c/d is 2011-03-04, but the wind speed/direction panel looks identical, so which date is correct? Figure 5: temperature and salinity taken from 10_4 and 11_5 buoys: is this respectively, or are the lines on the plot an average of them? The panel letters (a,b,c etc) are inside the figure in Fig 5 but outside of the panels in all other plots (outside the plot is clearer in my opinion, but as long as they are consistent, whichever is fine). Fig 5 caption: 'Hydrographic changes during the events in winter of 2010 are shown in Fig 4'. The dates in Figure 4 are for 2011 and 2013.

Discussion, Ln 244: Is this 18% underestimation compared to the DMI station? Perhaps make that clearer if so, with 'underestimates downslope flow by XX%, compared to observations at DMI, for an...'

Ln 317: Here you say that a typical DWE removed the sea ice: is this statement related to the March 2011 event from Figures 7+8? If so, I wouldn't call this event a 'typical'

[Figure]

DWE, as this was over 20m/s winds, and therefore an 'extreme' case by your definition. If this statement isn't related to the March event, where is your evidence for sea ice remove during typical events?

Ln 356/357: you do not mention or analyse the surface melt water or surface ablation in the manuscript, so how are you able to assume that there is no melt water or ablation? Is this purely because it is in the winter? There are instances of winter melting in Greenland and Antarctica, however. Perhaps reword this sentence to make it clear that your inferences are from winter characteristics as opposed to actually analysing the surface conditions.

Ln 370-373: I don't quite understand this sentence- please reword or restructure to make it clear.

Technical comments: Abstract, Ln 14: change 'impact on glaciers stability' to 'impact on the stability of glaciers'. Abstract, Ln 18: 'dependant' should be 'dependent'. Intro, Ln 27: full stop missing after Mouginot citation. Intro, Ln 27: 'calving on' should be 'calving of'. Intro, Ln 61/62: Put all citations at the end of the sentence to make it easier to read. Data/Method, Ln 116: 'Temperature profiles for each site was...' should be 'temperature profiles for each site were...' Data/Method, Ln 117: ADCP has not been defined. Results, Ln 158: 'had' should be 'have' Throughout: Be consistent with use of FS or Fjord Station for the AWS. Throughout: Point to specific panels in Figures where necessary to aid understanding. E.g line 158 is highlighting Figure 2a, whereas line 160 is Figure 2b. This becomes quite important for Figure 3,4 and 5. Ln 200: 'subsequence' should be 'subsequent'. Ln 232: 'the prevalence of DWEs was...' to 'the prevalence of DWEs are...'. Check your tenses throughout, as there are other instances where present tense would be better. Ln 255: singular/plural issue here: 'During a DWE' or 'During DWEs'. Ln 262: Katabatic is with capital K here- check consistency throughout. Ln 321: should facilitates be facilitated? I think this sentence should be re-worded to make clearer. Ln 339: 'on' should be 'from' Ln 346: 'retreat' is missing after 'rapid' Ln 354/355: 'indirectly' and 'directly' should be 'indirect' and

'direct'. Ln 361 to 364: A long sentence without punctuation to break it up- perhaps re-write into some shorter sentences, which would also make it clearer to understand your main point. Ln 415: 'the' should be 'there', and remove 'in' after trends.
* * *

---

## Referee Comment (RC2) · Anonymous Referee #2 · 21 Sep 2020

General comments:

The authors state a great working hypothesis as the title of their manuscript. As such the manuscript gives new insights into glacial retreat triggered by katabatic wind events. However, the manuscript needs to be significantly improved in order to convincingly test the working hypothesis, which is novel, and conclude their findings. In general, I think this is very important work, and will be of broad interest to the geoscience community.

Scientific quality:

Although the title reflects the content of the paper, the working hypothesis should be much better included (as a research question) throughout the entire manuscript; specif-

ically in the end of the introduction, the discussion, and conclusion. In general, I believe that large parts of the manuscript need to be clarified and reorganized as described in more detail below. Literature is mostly cited appropriately, however the new contribution from this study needs to be pointed out more appropriately.

Presentation quality:

Although the manuscript is well-structured, I would prefer, if it would be restructured following a research question (i.e., the hypothesis stated in the title). Figures are of good quality. Scientific English language needs to be improved, i.e., many statements need to be more precise and a number of very long nested sentences should be split in two or three. I also suggest to define terms like ice mélange and katabatic wind once and then stick to those terms instead of varying expressions (e.g., katabatic flows, increased winds, etc).

Structure and content of the manuscript:

I suggest the following reorganization of the manuscript to test the hypothesis stated in the title. First, it needs to be shown that observations suggest a direct link between katabatic wind events and glacier retreat (i.e., showing & analyzing all 3 (not only one) identified events that were found to be linked to a major glacier retreat, also giving numbers for how much the glacier retreated and the respective timing). Afterwards, you should aim to analyze the relevant mechanisms explaining this link. In the manuscript, I find two dominant mechanisms that are analysed in detail:

(i) Katabatic winds remove sea ice from the glacier front thereby changing the buttressing effect of the ice mélange on the glacier (i.e. preventing calving). As a consequence, the glacier speed increases (can this be shown?) and I suspect that the glacier front advances, subsequently leading to calving of icebergs. If enhanced calving can lead to the observed retreat of Helheim glacier by 1.5 km remains unclear to me. Can simple models help to answer this question?

[Figure]

(ii) Katabatic winds enhance the ocean circulation causing an increased supply of oceanic heat into the fjord. An enhanced ocean circulation (i.e., melt water/heat is transported more efficiently away from/toward the glacier) leads to more efficient melting at the glacier front. As such, enhanced submarine melting and undercutting can trigger a glacier retreat. Again, it remains unclear to me how large an increase in melt rates would need to be to trigger the observed glacier retreat of 1.5 km.

Ideally, I would like to see that both iceberg calving (due to the break up of sea ice) and enhanced submarine melting (due to an increased ocean heat supply) are analysed with respect to their potential in driving the observed retreat. Simple assumptions/models may help to quantify this (i.e., giving numbers instead of speculating only).

Methods and analyses of atmospheric data:

I really appreciate your definition of DWE. However, in order to get a better idea of how well the meteorological observations compare to ERA-5, I would appreciate more statistics to be presented (maybe even showing timeseries of wind speeds). How high is the correlation of the timeseries in wind speed? Is the correlaion statistically significant? Furthermore, I am surprised of your choice of the ERA5 pixel. I wonder if wind speeds from the fjord mouth or head are larger in ERA-5. I expect katabatic winds to flow from the ice sheet down the tidewater glacier and along the fjord, i.e., channelized by the topography. Thus, it does not seem plausible to me to pick a location outside the fjord some kilometers away from the fjord entry where due to the topography, the effect of katabatic winds can be expected to be much reduced and winds to be redirected (compared to katabatic winds coming down the glacier). The only reason to pick this pixel is the existence of a meteorological weather station there. Please discuss sources of errors and your choice of location in more detail with respect to the topography. You state that you do not find a long-term trend in the amount of katabatic wind events per year. However, did you also check if there is a long-term trend in the intensity of DWE's? And/or in the ice thickness/ice concentration? Both could imply that the mechanisms you are studying that drive glacial retreat may enhance in the

future. Thus, I find this worth to discuss.

Analyses of ocean data:

The authors refer to previous studies that showed and analyzed the same ocean data (mainly Jackson et al, 2014 and Spall et al, 2017). Still, the "background" fjord circulation and water masses are not explained sufficiently here and partly incorrect. It might be beneficial to include a physical oceanographer as a co-author for improved understanding and interpretation of the ocean processes.

Spall et al, 2017 give a nice summary of the water masses and the circulation in the fjord (based on previous studies). Furthermore, they study the effect of katabatic winds on the ocean circulation using Era Interim (in contrast to Era-5) and mid-fjord mooring data (same as here, I suppose) in comparison to their modeled circulation. They nicely generalize the fjord observations based on 8 katabatic wind events. Their results of changes in the fjord circulation related to katabatic wind events (Figure 9 in Spall et al. 2017) look similar to what is shown in this manuscript (Figure 4). What is new in this manuscript is that also hydrographic data and data from other mooring positions are included. This needs to be pointed out!

In the introduction, please introduce the relevant water masses (Atlantic Water (AW), Polar Water (PW)) and glacier-ocean interaction processes (subglacial runoff, submarine melting, meltwater plumes). Furthermore, I suggest to directly compare ocean velocity with hydrographic data, e.g., by including isotherms or isopycnals in the velocity time series (using the same DWE's).

The interpretation of changes in the ocean circulation is partly wrong and misleading. Please study Spall et al., 2017 who nicely explain the dynamic processes driving the observed changes (- I like to mention that I am not a co-author of that study -). I suggest to define 3 regimes to explain the changes on the ocean circulation, calving, and glacier retreat in response to katabatic wind events in a sketch (e.g., with a sketch design similar to Straneo et al 2010, Nature Geoscience, Suppl. Fig 1):

- Regime 1, "Background fjord circulation": AW inflow at depth, PW (and surface runoff) exported at (sub)surface; melting at the glacier front (water is always warmer than the local freezing point of seawater) causing less dense water to rise and to mix with ambient water etc.

- Regime 2, "Onset of katabatic winds": Wind stress enhances the export of PW (in the Ekman layer), to maintain the mass budget the AW inflow increases as well, this enhances the fjord circulation and leads to more efficient melting at the glacier front which in turn can cause undercutting. A pressure gradient that will balance the wind stress builds up (see Spall et al., 2017). This goes along with an uplift of isopycnals/isotherms at the calving front. It will depend on the wind speed how long it will take to build up a pressure gradient that compensates the wind stress and the reversed circulation (regime 3) sets on.

- Regime 3, "Reversed circulation": The pressure gradient (low pressure at fjord head, high pressure at fjord mouth related to sea surface height changes) will cause a reversing surface flow into the fjord that must be compensated by on outflow of warm AW at depth. Undercutting would be most efficient in Regime 2 and may contribute to increased calving. I do not claim this to be true, but this is at least my understanding after reading your manuscript and studying the related literature. I would like to encourage you to study these ocean dynamics in more detail. I believe that a sketch describing these ocean dynamics in relation to changes in the sea ice cover and calving would be a very nice summary of your findings.

Furthermore, instead of showing similar velocity fields as in Spall et al., 2017, you could take this one step further and generalize typical velocity/temperature profiles related to the regime shifts explained above. To point out the regime shifts during a DWE, I suggest to show only one event exemplarily (e.g., 15.03.). To see the changes in the temperature gradient (i.e., lifting isotherm/isopycnal), it would be nice to compare profiles (not timeseries, maybe using averaged profiles from different events) from all 3 regimes with each other. A Figure showing temperatures and velocity profiles could

be very nice to explain the different stages of the ocean circulation/properties.

Furthermore, I would make more use of all the other mooring data. You could try to combine data from all 4 locations (fjord head, mid-fjord, fjord entry/mouth, shelf) at one point in time (e.g. for regime 2, interpolating a section of ocean velocities/temperatures along the fjord) hopefully seeing sloping isotherms/isopycnals along the fjord. I really appreciate that you try to quantify changes in submarine melting caused by changes in the ocean heat content. You refer to Jackson et al., 2014, however, having a look to their method chapter, I find that SMR = (T-Tf)2. I believe that what you define as SMR is already the PDS. Please double check and clarify!

Discussion and Conclusion:

In the discussion and conclusion, you basically repeat what you showed in your results. This needs to be rewritten focusing more on the objectives of a discussion/conclusion chapter.

The discussion should primarily focus on interpreting your own results in relation to state-of-the-art. What can be generalized from your results? What are unexpected results (discuss potential reasons)? What are methodological limitations? What is new compared to previous studies? At which other glaciers this could play a role (considering not only Greenland but also Antarctica)? Any suggestions for wider implications (I am thinking of increased ice discharge from the Greenland ice sheet and all its implications)? Furthermore, it is good practice to discuss your (best) key result first.

Consequently, I would suggest to not use similar subsections as in your result section but focus on the link you find between katabatic winds and glacier retreat and discuss the relevant processes (break-up of sea ice & increased calving / increased inflow of warm Atlantic Water & enhanced submarine melting/undercutting). Please resort and rewrite accordingly.

The conclusion should sum up the advances of knowledge that emerges from your

paper. I find your conclusion a way too long since it includes more a summary of your paper and some discussion that should be moved accordingly. Instead, you should review your hypothesis (research question) here. Repeat your overall objective in one sentence. What is the overall take-home message that you want to tell the reader? Write down your learning message.

Minor comments:

L 14 & 15: Both sentences start with "Using". Rephrase.

L 15-16.: Along the fjord is from the fjord head to the fjord mouth (along its length) while across the fjord means in your case an East-West cross-section (across its width). The oceanographic measurements (not only hydrographic (T/S) but also ocean velocities are analysed!) were taken at different positions along the fjord. Please rephrase accordingly (not only here but also further down).

L 16: "during individual katabatic flows": I advise you to stick to one expression, namely "katabatic wind events" throughout the entire manuscript.

L 16: "Changes in mélange presence" – Please be more precise and add "ice". You could also write changes in the "sea ice cover/extent" alternatively.

L 20: Since you showed that the number of katabatic wind events did not increase over the last decades, is there another related property that changed? Wind speed? Temperatures during katabatic wind events? Reduced sea ice cover/thickness in the fjord?

L 22: "causing a retreat of up to 1.5 km". – Please be more precise, e.g., "causing a grounding line retreat by 1.5 km" (if appropriate)

L 23: "indirect influence on glaciers" – Isn't the impact you describe also indirect? It is not the katabatic wind that directly forces the glacier to retreat but it is via changes in the ice cover (leading to increased calving) and the ocean circulation (leading to increased basal melting(undercutting) that the glacier retreats.

L 24: "downslope wind events": see my comment above. Stick to one expression, i.e., "katabatic wind events".

L 27: "On average . . ." – This sentence is quite long and information seems to be a bit mixed up. Furthermore, please use more general citations of Greenland wide studies as given, e.g., in the introduction in Schaffer et al., 2020, Nature Geoscience ("The two major drivers attributed to the mass loss are increased surface melt caused by atmospheric warming and an increased ice discharge due to the speed-up of marine-terminating glaciers and ice streams. Oceanic heat fluxes causing increased subma-rine melting have been shown to be a dominant driver for the glaciers' speed-up and retreat.")

L 32: "discharge of cold and fresh meltwater" – please be more precise and add "at the underside of glaciers" or sth. The meltwater you are talking about (stemming either from submarine melt or subglacial runoff, i.e., surface melt that drains through the ice sheet to its underside and enters the ocean at the grounding line of glaciers) enters the fjords at depth (i.e., different from surface runoff)!

L 36-49: I am missing an introduction to the water masses, i.e., Atlantic Water (related to oceanic heat that drives submarine melting of Greenlandic glaciers) and Polar Water, which is essential for your work.

L 36: "Circulation . . ." – Not clear, please rephrase. E.g., "The glacier fjord circulation is governed by transport of oceanic heat at depth toward the glaciers and an export of fresh glacial meltwater (and Polar Water) at the subsurface to the continental shelves."

L. 38: "impacts both to the Greenland Ice Sheet and global circulation" – Please be more precise. Also add "e.g." before your citations.

L 39: delete "however". You could also rephrase to "A number of studies investigated the impact of barrier winds in driving inflows of warm waters of Atlantic origin from the continental shelf into glacial fjords (citations)." To be more precise.

L 40: "which are" – start a new sentence

L 42: delete "the associated"

L 43: "colder hydrographic conditions inside fjords" – at depths or surface? Please specify that this refers to Atlantic waters found at depths that carry heat toward the glaciers (- at least this is what I suppose you refer to, please double-check).

L 44: "often known" – rephrase "referred to as" or sth similar

L 45: "Atlantic Water" – here you write Atlantic Water (AW) for the first time. You need to introduce it well before (with its properties, see e.g. Straneo et al. 2012, Annals of Glaciology).

L 46: "fjords": more precise, you refer to glacial fjords at the southwestern coast of Greenland, right?

L 46-49: - more precise "potential sources" for what? - You could easily split the sentence in 2. - Are there studies on Antarctic ice shelves/glaciers that could be compared to/cited? - Why don't you introduce more what has been shown by Spall et al, 2017 already in more detail? Your study nicely follows up on their results! They give a nice introduction to the water masses and already discuss the circulation changes due to katabatic wind forcing in Sermilik Fjord. Please add their main findings to your introduction and use it as a motivation for your work!

L. 51: "in our case". Well, it's not your case. It is the case of the Greenland Ice Sheet.

L. 52: "above sea level": I suggest to add "toward the coast"

L 54: "in site temperatures" – where? At the ground? All along the glacier/ice sheet/fjords?

L 54-55 "This type of air flow is" – replace by "Katabatic winds are"

L 56 "Although" – I would delete "although"

L 58-59: "more intense" – could one give numbers (e.g., double as intense, or 5 times stronger, or sth appropriate)?

L 61-66: Please move this to the paragraph above where the fjord oceanography is introduced. Furthermore, you should separate a background circulation/stratification from the effect of katabatic winds. Even without katabatic winds, we find the circulation you describe in L. 61-62, i.e., AW flowing toward the glacier (driving melting at the calving front) and a return flow/export of PW and meltwater (or better glacially modified AW/PW) out of the fjord.

L 63-66: I suggest to delete the 2 sentences on the intermediary circulation. You can use it for your discussion, which you do anyway.

L 66-69. This sentence is not clearly written. Please make 2-3 sentences out of it. Explain in more depth how fast ice/an ice mélange impacts on calving and add more references. Since this is very relevant background for your analyses, you could have a whole paragraph on the role of ice mélange/fast ice on glaciers including recently observed changes in the ice cover around Greenland and the role of winds and air temperatures in driving these changes. Also, please define what "ice mélange" means.

L 70: replace "within the marine-terminating glacier system" by "on glacial retreat" or sth similar

L 72: replace "a relative abundance" by "the availability"

L 75: "katabatic winds" – I suggest to use the term "katabatic wind events" (see above)

L 77: Please refer to your hypothesis stated in the title! What are you aiming at? Please make use of your nice title by stating your hypothesis, why this is relevant, and how you want to test it.

L 79: I would appreciate more precise titles. Sth like "Wind data from weather stations and ERA-5", accordingly for the next subsections.

L 80: Rephrase. "We analyse data from two weather stations recording meteorological data in Sermilik fjord and its vicinity."

L 80-81: The DMI station is not placed in Sermilik fjord but outside. This needs to be mentioned and resulting limitations discussed. Please be more precise – where exactly is the weather station? How well may it capture katabatic winds? What is the temporal resolution of wind data? Accordingly, for the second weather station.

L 82: "banks" – please be more precise (eastern coast close to the fjord entrance, x km away from the fjord head)

L 80-84: You use the verb "provide" in each sentence of this paragraph. Furthermore, it may be worth to mention here (or somewhere below) that the glacier is facing in East-West direction while the fjord is in North-South direction. Thus, downfjord wind measured at the weather stations means along topography. But is that true for the DMI station? How is the terrain? I expect a steep coast there. I would expect that reanalysis data from the fjord entry should better represent katabatic wind events.

L 87: "which is similar to previous work" – better rephrase, e.g., "following Oltmanns et al, 2014"

L 88-89: "If not..." – obsolete

L89: "a wind direction" – rephrase to e.g. "Furthermore, we defined a wind direction interval to seperate ..."

L 90: more precise – barrier winds, i.e., coast-parallel/perpendicular to katabatic winds

L 91-93. Make 2 sentences out of this one and be more precise ("outer" coast? There is a coast in the fjord as well). - "underestimation" – do you mean underrepresentation? - "relative to" ...what has been observed (more precise).

L 94: replace "its" by "their"

L 95: But did you double-check that wind directions are reasonable? Wouldn't you

expect that katabatic winds are much more channelized inside the fjord and thus winds should be stronger inside compared to outside the fjord?

L 96-98: I am still puzzled by the wind direction. I believe that the topography is very steep there and thus I cannot imagine that katabatic winds flowing down Helheim glacier and Sermilik fjord will be measured at the DMI station since the topography would block the flow. Please explain/discuss.

L 98-99: Isn't that a result already? Would be nice to see timeseries of wind speed and directions in comparison.

L 97-109: Please start a new paragraph to introduce ERA5 data and resort your sentences. What is ERA5? Reference? How does it compare to data from the weather stations? (is that part of your results or was that already shown in Oltmanns et al 2014?) How well do the time series of wind speeds/directions correlate? It is great that ERA5 compares best at the chosen pixel to the weather stations since the DMI station is located inside the pixel. But (at least to my view) based on the topography one should use the pixel covering the fjord entry to study katabatic wind events (or at least both pixel).

L 107: "The creation of..." – rephrase "Based on our catalogue of katabatic wind events we picked/separated/studied changes in the ocean, sea ice and glacier state in Sermilik fjord during these events." or sth similar

L 108-109: last sentence needs to be shifted to further above

L 110: This is not only hydrographic (T/S) data but also current velocity data. Rephrase to e.g. "Moored ocean temperatures and velocities"

L 111: Buoys are not the same as moorings. Buoys are installed at the ocean surface or on sea ice drifting with the currents while moorings are moored at the seafloor and thus fixed to a position. You are not using buoy but mooring data. Please change throughout the entire manuscript. - "across" – you mean along, see comment above

[Figure]

L 112: "was" – data is plural; several citations for Straneo et al., 2015 should be referred to with 2015a-c accordingly in the reference list

L 113 "placed" – change to "deployed"

L 114: within the accession" – I do not understand. Please rephrase. The last sentence in this paragraph can be deleted. Simply refer to Fig 1 in the previous sentence. What is the setup of these moorings? In which depth are T/S loggers (SBE37?) and ADCPs installed? At which frequency (38/75/120/300kHz?) do ADCPs operate and thus cover which depth range? A table with mooring names, deployment/recovery time, instrument names, depth, temporal resolution would be useful.

L 116: "to give a better resolution of the water column, similar to the steps taken by" – replace by "following"

L 117-119: 'backwardly interpolated from the bottom of the profile up" – rephrase. You interpolate between points. How can that be backward? Or did you extrapolate? Are the ADCPs deployed close to the seafloor? How much of the water column do they cover?

L 119: "were created for three days either side" – rephrase, e.g., "were extracted lasting from 3 days before to 3 days after a DWE

L 120: The last sentence can either be skipped or more statistics need to be presented. How many events were relevant? How do you define relevant?

L123: You could start a new subsection here since this is a method related to ocean measurements. Rephrase to e.g. "We calculated ocean-inferred potential SMR assuming that the water column heat content is represented by our moored temperature measurements."

L 125: Rephrase to "SMR's were derived following Jackson et al 2014 from:"

L 127: In Jackson et al. 2014 this is already the PDS.

L 128-129: For which depth did you calculate Tf?

L 132-133: Do all stations cover the Atlantic Water core? If yes, I would suggest to use the maximum temperature instead of the mean temperature.

L 133-134: "Instead . . ." – I do not understand this sentence. Please rephrase.

L 136: It looks like you simply normalized the PDS. This does not need to be a new formular but can be written in the text.

L 140: Better start with what you want to use the data for, e.g., "In order to detect changes in the sea ice cover and glacier extent, we use satellite imagery . . ."

L 141: delete "MODIS . . .but only" and write that you used the AQUA 721 channel because . . . You could already mention here that it cannot see through clouds which is limiting your analyses.

L 149: Please start your sections with what you want to show or analyze next and why.

L 149-150. Split the sentence in 2. It is very confusing with so many commas.

L 151: "respectably" – you mean "respectively", I suppose. Lots of your wording should be improved. I am not a native English speaker but I would appreciate an improvement of your scientific English.

L 152: "as shown by . . ." – where is that shown? Not in Fig. 2, I suppose. . .

L 155: "Over the . . . timeseries" – please add the time span in brackets as a reminder

L 157: ". . . giving a total count of 199 events" please add "between year xxx and xxx"

L 158: "similar numbers" – don't you rather mean "counts" or "amount" or sth?

L 159: "the correspondence was extremely pronounced" – more precise.

L 161: "good agreement" - in what?

L 163: "significant relationship" - do you mean correlation?

L 164-165: "At the two…" – what does that mean? Is that for one event? I do not understand. Please rephrase and/or be more precise or skip.

L 167-172: As stated above I would suggest to define how the "normal" mean fjord circulation looks like (background) and then compare to what you find during katabatic wind events. The mean circulation during katabatic wind events was described based on the same (?) data by Spall et al. 2017. They do a very good job in describing the circulation, hydrography, and ocean dynamics playing a role in relation to katabatic winds. What is new here? I do not say that your work is not relevant but you need to make much clearer what is new and why it is interesting.

L 174-175: For which of the 2 events? Your color code is very difficult to read. I cannot see maximum speeds since you use the same colors for 0.3-0.75 m/s in b. Please choose more distinct colors. Furthermore, it would be nice to add isotherms (or isopycnal). Then you could directly relate e.g. to an Atlantic Water inflow.

L 175: "flow rate" – rephrase to "current speed/velocity" (better reserve the ford "flow" for winds)

L 176: "little noticeable" – numbers? Is that the same for each DWE?

L 177: "increased current speeds" – hardly seen in d, please change the color code

L 177: "wind stress" – did you mention wind stress before? You should certainly do that and introduce the relation between wind stress and the ocean. Does rotation play a role here? What is Rossby number?

L 179: "the upfjord current was not previously present" - Do you show that? in d there is an inflow but it is just small (and due to your white color code not well visible). Again, what is the normal state of the fjord circulation? Is there one? Could you show and discuss a background flow field before showing the effect of katabatic winds? Why do you show 2 examples? You can show one pronounced one and say that others look similar. Or as in Spall et al. 2017 use a mean field for katabatic wind events.

L 180: "525-550m" – How shallow is the sill at the fjord entry? Water deeper than the sill depth is presumably exchanged much slower. I would not show bottom velocities at all but simply state that the ocean close to the seafloor is very quiet. Maybe you can even refer to Jackson et al 2014 or Spall et al 2017 in case it was mentioned there.

In general, I suggest to use terms like "deep AW inflow" and "shallow PW outflow". But this is up to you.

L 185: "seems to be the upward movement of the deep-water current" – what do you mean by this? Again, the circulation was nicely explained in Spall et al 2017.

L 187-188: "The orgin..." – is explained in Spall et al. 2017. I recapture here: katabatic winds "push" surface waters out of the fjord. A pressure gradient builds up with low pressure in the inner fjord and high pressure at the mouth of the fjord. This goes along with lifting of isopycnals/isotherms at the inner fjord, i.e., warmer waters are lifted to shallower depth (potentially increasing melt at the glacier front). The pressure gradient will subsequently drive a surface current into the fjord (counteracting the previous export) until the "normal background field" is reached again. This is how I would interpret your observations and it is analogous to Spall et al 2017.

L 190-201: I would prefer to analyze one representative event or generate a mean temperature field that represent conditions during DWE's. You could also use isotherms on top of the velocity field (as stated above) and/or compare representative profiles from before, during and after a DWE. It would be fantastic to see how temperatures are changing all along the fjord since I would expect the thermocline to become shallower at the fjord head but deeper at the fjord mouth. You could e.g. pick the 3°C isotherm and compare its depth at all mooring sites for the different timings.

L 198: "Fig. 5" – c? Please be more precise what subfigure you refer to (in case you keep all of them what I would not suggest)! "a sharp temperature jump occurred" – at the surface or where? - Do maximum temperatures at shallower depth correspond to the timing of maximum outflow velocities?

L 199 "The peak" – more precise! E.g. "The increased temperatures associated with an increased import of AW (?) . . ."

L200 "Subsequence peaks. . ." – peaks in what? More precise,

L203 "showing water column heat content" – well, the SMR does not show the heat content but can be used as an indicator of the heat content. Please rephrase.

L 203: "across the fjord" – you again mean "along" I suppose. Furthermore, I would not expect to see the same signal outside the fjord compared to inside the fjord. Please differentiate and explain!

L 208: "100% increase" - in what?

L 210: "association" – do you mean correlation/stronger link?

L 209-210: "The maximum . . ." – Since event 4 shows by far the longest duration (and the relation does not need to be linear), maybe a combination of both the wind stress and how long the wind force acts on the sea ice/ocean surface is most relevant to drive the enhanced inflow of AW and thus increased submarine melting.

L 210: Please try to always link your sections. Here you could e.g. nicely link by sth like "Next, we will accordingly analyse changes in the sea ice cover in order to . . ."

L 215: "held within it were released" – held within what? Also "exported" might be a better word than "released".

L 216: "small section" – section? What do you refer to? Of what?

L217: Fig 7b (add b)

L 218: "full-thickness" – how do you know? What does it mean? Same thickness as calving front of Helheim glacier? How thick is it?

L 219-220: "Initially. . ." – Is that visible in your pictures? Please mark the icebergs you are relating to. Most interesting to me seems Fig 7a. There is so much open water that

the pressure on the sea ice and glacier will be released and the glacier can calve off more easily.

L 221: "the terminus ..." – This is pretty impressive and should be a real highlight in your paper!

L 223: "A similar pattern ... after strong events following" – Please be more precise! Pattern of what? Strong events of what? I suggest "We observed/found a similar retreat of the glacier front... strong DWE's triggering/driving the ..." In my view, this is the most interesting new finding of your study. Instead of showing several examples of how ocean velocities/temperatures are changing during DWE's (which was shown before), I would prefer to see all examples where you find a direct relation between DWE's and glacier retreat. And subsequently have a look to changes in ocean and sea ice conditions during those events.

L 225-226 (also lines 325-328): "Sea-ice ..." – I do not see that ice is moved to the right. Instead it seems that on the right-side (facing from Helheim glacier down-fjord) there is open water. Which suggests that ice was pushed away either to the left or down-fjord. Please discuss in general (as mentioned above) if rotation plays a role considering Rossby numbers and the width of the fjord (also done in Spall et al., 2017). Furthermore, you wrote earlier "sea ice" instead of "sea-ice". Please adjust.

L 226-227: "We also found ..." – At the southern end of the calving front you also have more open water which may be linked to increased calving and/or warmer ocean waters rising from depth subsequently leading to glacier retreat. It would be nice to see if this was also the case in the other 2 events in 2005 and 2013 and worth to discuss the potentially related processes driving glacier retreat in more detail.

L 232: "independently" – of what?

L233-235: reference?

L 237: "wind speed of 90 m/s" – where are those wind speeds observed? Further

inland?

L 238-239: "We attribute . . ." – please discuss the role of topography in more detail, e.g., is there any study showing by how much katabatic winds slow down downstream the ice sheet/glacier?

L 244: "by 18 +/- 6.9%" – why is that not part of your results? Furthermore, I suggest to split the sentence in 2 and rephrase it.

L 250: "model ocean heat loss. . ." – please give more examples why this has wider implications (e.g., can be applied to other glacier-ocean systems around Greenland etc)

L 253: "pycnocline" – you did not show any pycnoclines in Fig. 3. It would be lovely to see pycnoclines on top of the velocity field (if possible)!

L 252-257: It would be lovely to discuss a schematic of your 3 regimes: before, during, and after the katabatic wind event. You could not only include changes in the fjord circulation but also in the sea ice cover! That would be an amazing summary! I have sth similar in mind as Supplementary Fig. 1 in Straneo et al., Nature Geoscience, 2010 but of course adjusted to your findings.

L 259: "cold glacially modified water near the surface" – Again, I would prefer if you are more precise. Surface suggests like you are talking about the upper 10-20m, while instead you are talking about the upper about 200 m (which includes Polar Water).

L 263: "trend" – Rephrase, this is not a trend.

L264: "uniform no matter what the location" – Is that true? I would love to see a spatial distribution of temperatures/densities/velocities from your different mooring positions during one DWE. Is it possible to make a section/compare profiles from all locations? It would be interesting to see if one finds e.g. changes in the isoycnal slope along the fjord during a DWE.

L 268: "heat" – Ocean heat transport would be the integrate of the product of temperature and velocity fields across the fjord. It is worth to discuss limitations of your data set that do not allow you to directly compute heat transports. Furthermore, this infers that if only one property, i.e., either the temperature or the velocity increases, the heat transport would be increased. Here both temperatures and velocities are increased during DWE's (right?) leading to an enhanced fjord circulation and increased submarine melting at the glacier front (Worth considering also, how deep is the seafloor/how thick is the calving front? And how warm are maximum ocean temperatures at the grounding line?). Please discuss these aspects in more detail.

L 270-283: This paragraph can be shortened. You already introduced the intermediary circulation in your introduction. Here it would be interesting to compare increased heat transport/glacier retreat triggered by the intermediary circulation quantitatively to what you find for katabatic wind events (maybe over the course of one year).

L 289: "submarine melt rates" – I wonder if it is possible to estimate submarine melt rates in m/yr from either ocean heat transports or the glacier mass budget. However, I am not sure if data sets allow for that.

L 293-294: "that shelf-fjord heat ..." – I do not understand. Did you show this?

L 296: "potential SMR" – more precise. Give numbers.

L 296-305: Again, you need to explain the relation of wind speeds via wind stress acting on the ocean surface driving a transport of subsurface waters in the Ekman layer. It might be interesting to consider and discuss the effect of the drag coefficient that changes with sea ice concentration (e.g., Lükes and Birnbaum, 2005; Andreas et al., 2010; Ma et al, 2016). Thus, similar strong wind speeds (DWE's) do not equal a same magnitude in wind stress force acting on the sea ice/ocean surface.

L 308: "undercutting" – does it always need to be undercutting or can it also be that you get increased melting at all depths of the glacier front due to an enhanced circulation

that provides more heat going into melting the glacier?

L 310: "bergy bits" – what do you refer to? Please rephrase. In general, this sounds more like an introduction than a discussion.

L 317: "upper-fjord" – could be misinterpreted with respect to depth instead of horizontal location. Better use "inner fjord" or "fjord head".

L 318-320: "Simultaneous . . ." – This sentence is too long and very confusing. Please rephrase. Also, what do you mean by steadily maintained salinity? It is actually very interesting to see that salinities stay higher while temperatures already drop after a DWE. What is the effect in density? A T/S-diagram showing the evolution in time would be very nice to see and discuss. What could be the reason for the different behavior of T and S in time? Furthermore, you do not measure the surface temperatures. Thus, we cannot know if surface temperatures are at the freezing point or warmer and consequently melting can/cannot drive sea ice melt. However, in principle I also do not believe that the ocean is melting the sea ice from below because I suggest that meltwater plumes rising at the glacier front will detach already at depth (when reaching the density of the "background" properties).

L 321: Why don't you analyze/show air temperature changes related to your DWE's? Then this point would be stronger. However, I do not believe that air temperatures in winter/spring will be above Zero and able to drive melting. Instead, it rather seems that the fjord quite quickly refreezes (Fig. 7).

L 338: "temperature changes" – Please add "ocean". Throughout the whole manuscript you need to be more precise talking about ocean (not air) temperatures.

L 345: "no long-term trend in katabatic winds" – more precise. You only had a look to how often they occur per year, right? What about their strength? Did the intensity of DWE's increase? Did the ice thickness decrease? These would be interesting points to include in your analyses and/or discussion.

L 346: "the rapid of Helheim" – you are missing the word "retreat"

L 351: "showed retreat" – how far? In the order of 1.5 km?

L 356: "the absence of surface melt water as a driver of plume-induced undercutting" – I do not understand this. Do you refer to subglacial runoff here? However, the ocean will be always warm enough to drive melting and thus plumes rising at the glacier front, I suppose. However, they do not necessarily need to reach the ocean surface but detach once they have reached the density of the background stratification.

L 361-365: "Modelling studies .." – Please split this sentence in 2 or 3.

L 372-375: Can you quantify changes in the buttressing somehow?

L 399-400: "an absence..." - Did you discuss this? I assume that Jenkins, 2011 relates this seasonality to Antarctic glacier/ice shelf-ocean systems. Around Greenland, the ocean water will be always warm enough to drive melting and thus presumably meltwater plumes at the glacier front.

L 410: "in contrast" – I would not see it as a contrast! First, it is questionable to judge if the effect is "direct" or "indirect" (as mentioned above). I would rather tend to interpret the effect of katabatic winds in driving the glacier retreat also to be indirect since other intermediate processes play a major role. In principle, I would rather point out that next to wind-driven shelf-fjord ocean exchange flows also katabatic wind events can considerably change the ocean circulation triggering increased submarine melting and glacier retreat.

L 415 "the" – what? Missing word.

L 435: "Isbrse" – please check all your references for special letters (like æ) and capital letters!

Fig 1: Why don't you name sites 10_2 - 10_5 better 10_A-D according to the location to the fjord head, i.e., A=4, B=2, C=5, D=3. L 580: "Esri world imagery" – Is there a

reference?

Fig 2: Rename "Extreme Events" by "DWE > 20 m/s" or sth similar according to what is written in the Figure caption. In the Caption you do not need to write over which time period the records were taken since it is seen in (a) but rather point out the locations (according to Fig 1) and explain the abbreviation FS.

Fig 3: Why don't you use the same colorbar in b and d? In principle I like discrete colors but you use the same blue color for 3 different speed ranges. Why? Also, it is not intuitive that you use white color for velocities between 0-0.2 m/s in d (- these could easily misinterpreted as zero floe). This should better be colored in light red inferring up-fjord velocities. The data recorded at the seafloor (by what kind of instrument?) seems to be constant. Could be skipped.

Fig 4: It would be nice to show the same events as in Fig 3 to relate velocities to currents. Did you also record salinities at the same depths? Could you add isopycnals?

Fig 5: Same as in 4. Alternatively, it would be nice to plot a T/S-diagram as a scatter plot coloring the points with either time or wind speed or DWE regime 1-3.

Fig 6: What are the orange lines? I would love to know the timing of each event.

Fig 7: Do you need d-g? You only track the movement of the iceberg but not new calving after the 19th, right? Please encircle the icebergs with a colored line to make changes clearer.

Fig. 8: It would be optimal if the ticks of the x-axis in a and b are placed exactly on top of each other. Also, you could make the extraction smaller zooming in more to the calving front.

---

## Referee Comment (RC3) · Anonymous Referee #3 · 24 Sep 2020

The manuscript provides a basic count of strong wind events near a fjord with a tide-water glacier in south-east Greenland. The authors use data from Danish coastal wind stations and a coastal grid point of a data assimilation modeling (ERA5). The manuscript attempts to relate some of these strong wind events to fjord circulation and glacier melting, calving, and retreat. Raw time series of ocean current and temperature profiles and single-point salinity-temperature time series are shown a few days prior, during, and a few days after wind events.

The connection of the wind events to ocean circulation fails, because of an incomplete and uninspired discussion of raw data that are scattered in time (6 events) and space

(5 locations). There are many wiggles in the many plots, but what stands out is the mismatch of time scales and the confusion of what happens when and where and how this relates to all other events. I know that Drs. Rebecca Jackson and Fiamma Straneo worked on the fjord circulation using the same data with more dynamical insight. In contrast to the present authors Jackson and Straneo organized and synthesized data from different "event-types" more credibly via an ensemble averaging sense that defines a common time zero for each event. These published papers may serve as a model on how to properly process, filter, and average complex ocean data to provide statistical meaning and dynamical understanding.

All ocean data are noisy, that is, they contain high frequency spectral components that are not contained in the wind data. Some features appear to correlate with the wind data while others do not. The scales of all velocity (2 plots), temperature (4 plots), and temperature-salinity (4 plots) are all different from each other for each and all events. It almost feels as if the authors let MatLab do all the analysis via its automatic scale selection. This is very poor form and results in a cluttered and uninformed presentation. Some wind data series are presented more than once to perhaps demonstrate that this and that wiggle relate visually or not or with some lag or without a lag. I would not want my own graduate students to see and "learn" from such confused and chaotic show-and-tell lack-of-analysis.

There are many smaller items and problems that to me indicate a sloppy and rushed submission. Figure-6 refers to 6 events from 2010 that I could not find anywhere else in the manuscript. The use of the color "white" in Fig.-3 could indicate the absence of data (icebergs near surface perhaps?) or the velocity interval of 0.0-0.2 m/s (Fig.-3d) or 0.00-0.15 m/s (Fig.-3b). The author always refers to Fig.-3 (there is Fig.-3a through Fig.-3d) or Fig.-4 (there is a Fig.-4a through Fig.-4h). These are all minor quibbles, there are many more of these.

In summary, I recommend to reject the manuscript, because I could not find credible ocean data analysis or synthesis that adds value to the raw data. The wordy and

largely speculative manuscript fails to provide quantitative or dynamical understanding or even a unified view of how the many wind force ocean circulation and glacier melting. Reading this manuscript, I find that each wind event triggers a different ocean response. The presentation lacks analysis that would extract the katabatic wind effect from the ocean record. It compares a wiggle here with a wiggle there that may or may not mean much.